# Computer-aided discovery of connected metal-organic frameworks

Ohmin Kwon [1,3], Jin Yeong Kim [2,3], Sungbin Park [2], Jae Hwa Lee [2], Junsu Ha [2], Hyunsoo Park [1], Hoi Ri Moon [2] & Jihan Kim [1]

Composite metal-organic frameworks (MOFs) tend to possess complex interfaces that prevent facile and rational design. Here we present a joint computational/experimental workflow that screens thousands of MOFs and identifies the optimal MOF pairs that can seamlessly connect to one another by taking advantage of the fact that the metal nodes of one MOF can form coordination bonds with the linkers of the second MOF. Six MOF pairs (HKUST-1@MOF-5, HKUST-1@IRMOF-18, UiO-67@HKUST-1, PCN-68@MOF-5, UiO-66@MIL-88B(Fe) and UiO-67@MIL-88C(Fe)) yielded from our theoretical predictions were successfully synthesized, leading to clean single crystalline MOF@MOF, demonstrating the power of our joint workflow. Our work can serve as a starting point to accelerate the discovery of novel MOF composites that can potentially be used for many different applications.

[1] Department of Chemical and Biomolecular Engineering, Korea Advanced Institute of Science and Technology (KAIST), Daejeon 34141, Republic of Korea. [2] Department of Chemistry, School of Natural Science, Ulsan National Institute of Science and Technology (UNIST), Ulsan 44919, Republic of Korea. [3] These authors contribute equally: Ohmin Kwon, Jin Yeong Kim. Correspondence and requests for materials should be addressed to H.R.M. (email: hoirimoon@unist.ac.kr) or to J.K. (email: jihankim@kaist.ac.kr)

Composite metal-organic frameworks (MOFs) are comprised of one MOF and another material with noticeably different properties[1–3]. In general, MOFs are viewed as attractive candidates to construct new composite materials given their facile synthesis[4–7] and a large library of synthesized MOFs (over 70,000)[8] that can be used as building blocks. As such, many researchers have integrated MOFs with other classes of materials (e.g., other MOFs[9–15], carbon-based materials[16–18], oxides[19–22], metal nanoparticles[23–29], polymers[30–33]) to produce new structures with synergetic properties. Unfortunately, in many of these composite MOFs, the precise nature of the interaction and the bonding at the interface between the two materials is unknown and cannot be characterized well with any of the known methods. One major drawback that stems from this black box interface is the loss in tunability and control that prevents facile optimization of MOFs, given the nebulous nature of the interactions at the atomic and the molecular scales.

To remedy this drawback, one can envision a more rational approach where the materials are a priori designed with the molecular interactions between the two materials mapped out prior to their synthesis. One way to achieve this mapping is to use computational tools to identify the optimal pairings of MOFs that can facilitate the creation of the composites. As far as we know, no one has yet managed to synthesize composite MOFs based on theoretical predictions. Previously, Bristow et al. screened through various MOFs and matched these materials with different substrates to find new ideal pairings between the two classes of materials[34]. Similarly, Tarzia et al. used computational techniques to investigate MOFs that can be grown onto the $Cu(OH)_2$ substrate using microscopic level analysis[35]. In both of these theoretical works, the predicted materials from the computational analysis have yet to be synthesized. More recently, Zhou group reported several interesting works for synthesis of hierarchical MOFs that are composed of two different MOFs by kinetic control such as surface functionalization and temperature adjustment[15,36,37]. However, as far as we know, these composites were not a priori designed at the molecular level that take into account specific interactions between the linkers and the metal nodes of the participating MOFs.

In this work, a joint computational/experimental approach was adopted where we started with the hypothesis that a metal node of one MOF can coordinately bond with the linker of a different MOF; and one can intuit that the precisely matched interface configurations at the atomic/molecular level can enhance the likelihood of synthesizing MOF@MOFs, which are composite MOFs where a MOF is grown on a different MOF[9]. Subsequently, our newly developed computational algorithm identified hundreds of MOF pairs, of which six predicted pairs were successfully synthesized experimentally, with facet-oriented growth of constituent MOF on the surface of a pre-existing MOF being observed. As such, we posit that our workflow can enhance the likelihood of synthesizing MOF@MOFs in the form of large single crystals, and thereby demonstrate the utility of rationally designing the MOF@MOFs.

## Results

**Computational algorithm to identify MOF@MOF pairs.** In devising our MOF@MOFs generation algorithm, two main assumptions were made to determine whether two different MOFs can seamlessly connect together into a single crystalline MOF@MOF at the interface. First, there will inevitably be many defective sites preventing the MOF@MOF configurations if the metal nodes of the first MOF and the linkers of the second MOF are misaligned at the interface. Thus, the initial assumption is that the lattice parameters between the two MOFs should be nearly

identical (or multiples to one another), which leads us to consider only crystallographically linkable pairs. With this constraint being satisfied, the second assumption is that the interface possesses well-matched chemical connection points, which are defined to be spatial locations where linker/metal node of one MOF meet with the metal node/linker of the second MOF through coordination bonding.

To simplify analysis, MOF-5 was chosen as a first target MOF (Fig. 1) in our screening process, given its ubiquitous nature and facile synthesis. From the CSD MOF Subset database[8], MOF-5 (refcode: EDUSIF)[4] was computationally cleaved to expose the square (001) and the rhombus (111) surfaces. Amongst the infinitely many surfaces, these two surfaces were specified as targets for screening given that these are commonly exposed surfaces upon experimental synthesis and that MOF-5 can be cut along these two surface directions without breaking the intra-molecular bonding of the organic linker or the metal cluster moiety. Next, the 89,484 experimentally synthesized MOF structures from the MOF database was used in the initial screening procedure to find potential matching pairs with the MOF-5 structure. From this set, only the cubic (1976), hexagonal (4483), and tetragonal (3884) unit cells were extracted as candidate MOFs to simplify the lattice-parameter matching procedure. The three-dimensional unit cells of the remaining 10,343 MOFs were cleaved along few common surface directions (i.e., square cubic (001), square tetragonal (001), rhombus cubic (111), rhombus hexagonal (001)) using the Materials Studio package (Supplementary Note 1 and Supplementary Fig. 1).

In the lattice-parameter matching step, two dimensional lattice parameters of the cleaved MOFs obtained from the previous step were compared with that of the MOF-5 lattice parameters (i.e., 25.832 Å × 25.832 Å for the (001) surface and 18.266 Å × 18.266 Å for the (111) surface). Our algorithm filtered pairs of MOFs that possessed the ratio of the lattice parameters to be 1.0, 2.0, 3.0 (with 3% error threshold) to determine lattice-parameter matching[38].

For the local chemical connection matching procedure, our algorithm simplified the cleaved surface to contain only the atoms that can engage in coordination bonding at the interface between the two MOFs on the same plane (Fig. 1). For all of the candidate MOFs, the oxygen atoms of the carboxylic acid and all of the metal atom types from the metal clusters were selected as potential chemical connection points (Supplementary Fig. 2). To determine a proper match, the unit cells of the two MOFs were positioned to find the minimal separation between the chemical connection points (Supplementary Note 1 and Supplementary Fig. 3).

From the procedure outlined above, 86.0% of the tested MOFs were ruled out during the lattice-parameter matching procedure. One noteworthy MOF that cannot connect with MOF-5 was UiO-66; as can be seen from the chemical connection points for both the MOF-5 and the UiO-66 structures (Fig. 1a), the mismatch in the lattice parameters (19.9% mismatch; UiO-66 (RUBTAK): 20.7004 Å (001) and MOF-5 (EDUSIF): 25.832 Å (001)) leads to suboptimal matching between the two sets (Fig. 1a right) and as such, UiO-66 was one of many structures eliminated from our screening algorithm.

On the other hand, few hundreds of structures from the CSD database were found to possess compatible chemical connections points with MOF-5 (Supplementary Note 2 and Supplementary Data 1). Amongst them, HKUST-1[39] was identified to be a potential match despite the fact that these two MOFs possess completely different metal clusters ($Zn_4O$ cluster vs. Cu paddlewheel cluster) and organic linkers (1,4-benzenedicarboxylate (bdc) vs. benzene-1,3,5-tricarboxylate (btc)). Despite the

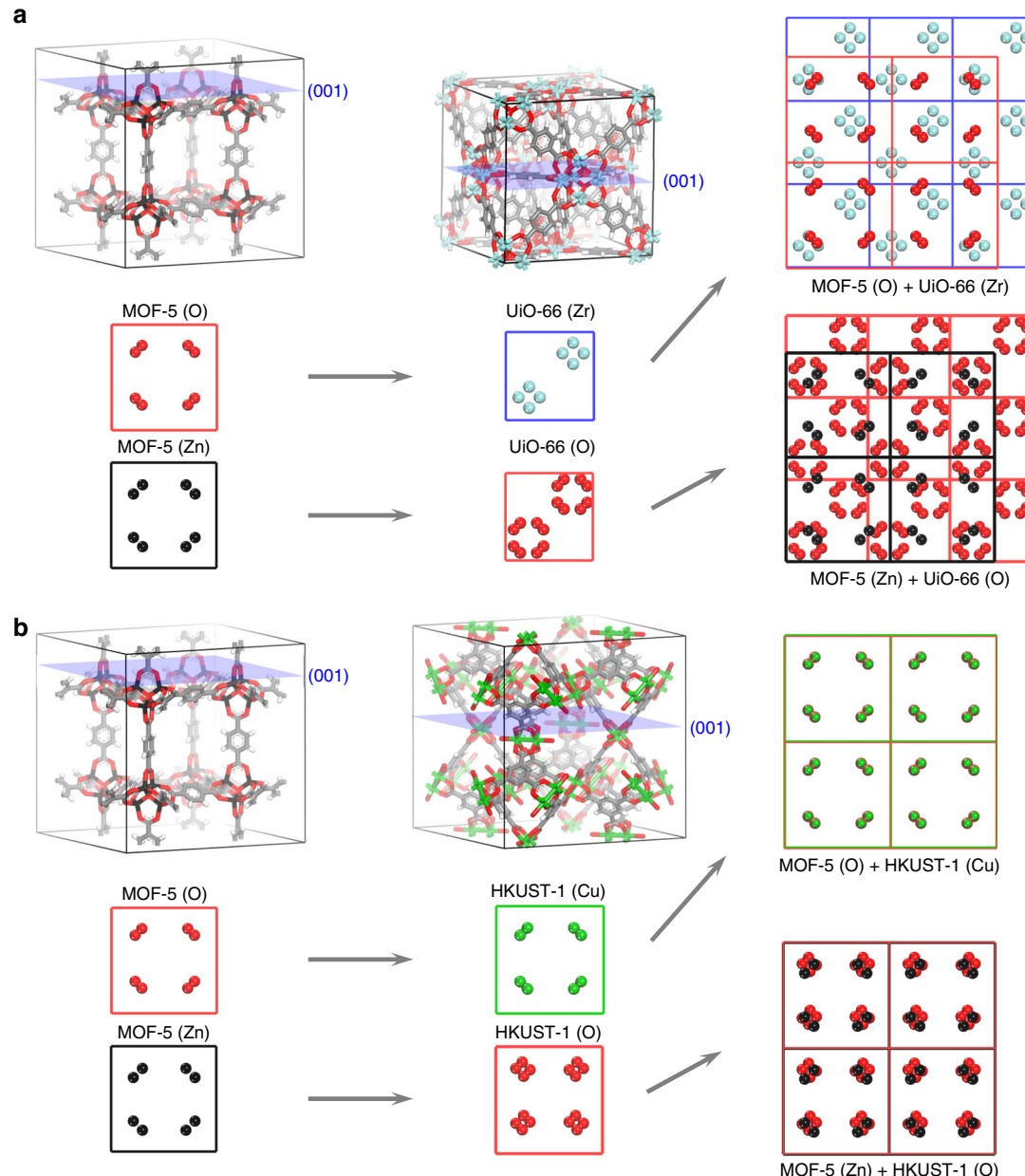

**Fig. 1** Illustrative example showing the chemical connection points matching. **a** Examples of MOF-5 and UiO-66 along the (001) plane. Black, blue, and red circles represent the zinc, zirconium, and oxygen atoms, respectively. **b** Similar to **a** but for HKUST-1 with green circle representing the copper atoms

differences, these two MOFs happen to possess very similar 2D lattice parameters (1.92% mismatch; HKUST-1 (XAMDUM): 26.3368 Å (001) and 18.6229 Å (111), MOF-5 (EDUSIF): 25.832 Å (001) and 18.266 Å (111)) in both the (001) and the (111) cases. During the screening process, the chemical connection points of HKUST-1 were generated along the (001) plane (Fig. 1b) and (111) plane (Supplementary Fig. 5). And the zinc atoms of MOF-5 and the oxygen atoms of HKUST-1 aligned nearly on top of one another along the (001) plane, and similarly the copper atoms of HKUST-1 and the oxygen atoms of MOF-5 aligned on top of one another along the (001) plane (Fig. 1b right). In a similar manner, the zinc atoms of the MOF-5 metal cluster and the oxygen atoms of the HKUST-1 btc ligand were separated by 4.10 Å along the (111) plane and the copper atoms of the HKUST-1 Cu paddlewheel were 2.88 Å away from the oxygen atoms from MOF-5 bdc ligand along the (111) plane (Supplementary Fig. 5).

From Fig. 1b and Supplementary Fig. 5, it can be seen that four different combinations of chemical connection points can match between MOF-5 and HKUST-1: (1) MOF-5(Zn)/HKUST-1(O) along (001) plane, (2) MOF-5(O)/HKUST-1(Cu) along (001) plane, (3) MOF-5(Zn)/HKUST-1(O) along (111) plane, and (4) MOF-5(O)/HKUST-1(Cu) along (111) plane. Given the aforementioned options, four distinct computational structures can be formed for each of the cases (Supplementary Note 3 and Supplementary Figs. 6–9). In comparing between (1) and (2), although differences exist at the interface between the MOF-5 and the HKUST-1 structure (with (2) being surmised as more favorable due to larger degree of freedom from the bdc linker compared to the btc linker), within the computational model, the atomic positions away from the surface are exactly the same for both (1) and (2) (Supplementary Figs. 6, 7). And similar argument holds for (3) and (4) (Supplementary Figs. 8, 9). As such, although the more feasible models were proposed in each of

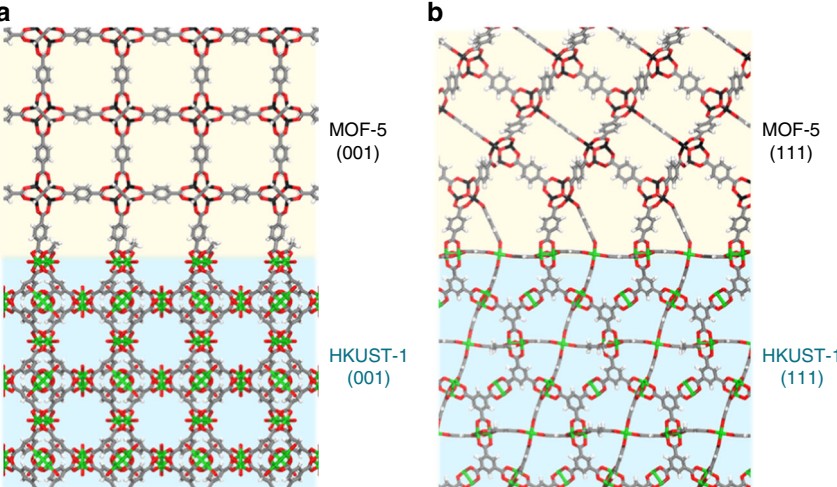

**Fig. 2** Computational structural models for the HKUST-1/MOF-5 system. **a** HKUST-1(001)/MOF-5(001) and **b** HKUST-1(111)/MOF-5(111). The yellow region indicates the space for MOF-5 phase and green region for the HKUST-1 phase

the (001) and the (111) cases with the copper metal atoms being connected to the bdc linker in each of the cases (Fig. 2a, b and Supplementary Figs. 6–9), we deem that the macroscopic phases will remain the same regardless of the models.

To test for the stability of our proposed HKUST-1/MOF-5 structures (Fig. 2a, b), density functional theory (DFT) simulations were conducted to obtain both the energy penalty incurred from the linker strain and the energy stabilization from the bond formations (Supplementary Note 4). The energy stabilization of bonding at the (111) surface ($E_{bonding(111)}$) was computed to be $-0.0229$ eV Å$^{-2}$ and the energy penalties from the strains in MOF-5 and HKUST-1 were $+0.0062$ eV Å$^{-2}$ and $+0.0025$ eV Å$^{-2}$, respectively. As a result, the net energy stabilization was $-0.0142$ eV Å$^{-2}$ at the (111) surface. Similarly, energy analysis for the (001) case was conducted and the net energy stabilization at the (001) surface was computed to be $-0.0060$ eV Å$^{-2}$ (Supplementary Table 2 and Supplementary Figs. 10–11). Given the relatively low energies of these structures, the HKUST-1/MOF-5 system was targeted for the initial experimental synthesis.

**Experimental synthesis of HKUST-1@MOF-5.** As a 3D substrate for the epitaxial growth of MOF-5, the octahedral-shaped HKUST-1 crystals (with {111} surface being mainly exposed) were initially prepared due to their stronger crystal stability against the sequential solvothermal reactions (Fig. 3a), which also has been exploited in other MOF@MOF syntheses[36,40]. They were heated at 85 °C for 36 h in the MOF-5 precursor solution (detailed synthetic routes in Supplementary Methods). As shown in Fig. 3b and Supplementary Figs. 12, 13, single crystalline HKUST-1@MOF-5 core-shell crystals were purely synthesized, with the blue HKUST-1 single crystals located within the center of the colorless cubic MOF-5 single crystals with seamless interfaces. Similar composite has been synthesized in the previous work but achieved via different approach of kinetic control of shell MOF growth[36]. The major morphology of HKUST-1@MOF-5 indicates that crystal orientation of a MOF-5 single crystal on the {111} planes of the octahedral HKUST-1 shows the [111] growth direction, as evidenced by the X-ray powder diffraction (XRPD) data (Supplementary Fig. 14), which is in good agreement between the simulated prediction and experimental results at the interface between the {111} plane of MOF-5 and the {111} plane of HKUST-1. The core and shell crystals obtained by breaking the HKUST-1@MOF-5 crystal were respectively

confirmed by single-crystal X-ray diffraction (SCD) analysis (as HKUST-1 for the core and MOF-5 for the shell; see Supplementary Fig. 15). The HKUST-1@MOF-5 crystal has a reasonable surface area of 3356 m$^2$ g$^{-1}$, considering the surface area values for each of the MOFs (i.e., 2021 m$^2$ g$^{-1}$ for HKUST-1 and 3598 m$^2$ g$^{-1}$ for MOF-5) (Supplementary Fig. 16). It should be noted that the color change of HKUST-1 core crystal (green to dark blue) occurs upon the solvent exchange of diethylformamide (DEF) into dichloromethane (Supplementary Fig. 17). This indicates that even after construction of the MOF@MOF structure, there exists molecules-accessible diffusion path at the interface between the HKUST-1 and the MOF-5 materials.

To elucidate the epitaxial growth mechanism of MOF-5 on the HKUST-1 surface, the growth process was monitored at different times (Fig. 3c and Supplementary Figs. 18, 19). Starting from 26 h of reaction time, colorless crystalline islands began to appear on the HKUST-1 surface. After further progression (28–30 h), the islands of MOF-5 continued to grow and merged into a larger single crystal as opposed to forming polycrystalline domains, and we posit that this is aided by the matching chemical connection points as predicted by the computational algorithm. Finally, at 36 h, the shell MOF-5 covers the entire surface of HKUST-1 in a core-shell structure by further growing with the [111] orientation of single crystalline cubic crystals. Next, given that the computational algorithm predicted match in along the (001) planes as well, additional experiments were conducted with the HKUST-1 crystals that majorly exposes the (001) planes[41] (Fig. 3d). Using the same synthetic conditions, the cubic HKUST-1 substrate was gradually coated with the shell MOF-5 crystal, along the growth direction of [001] (not [111]), thereby creating the cubic HKUST-1@MOF-5 structure in the single crystal phase (Fig. 3d and Supplementary Figs. 20–23).

To validate the predictions from the computational algorithm, it is noteworthy to attempt pairs of MOFs that lack the matching of chemical connection points, and as such, two different pairs (i.e., Co-bdc/MOF-5, HKUST-1/IRMOF-20) were selected as subjects. For Co-bdc/MOF-5, the attempts made to grow MOF-5 on the surface of Co-bdc led to polycrystalline crystals (Supplementary Fig. 24), in which Co-bdc has different crystal system (triclinic $P\bar{1}$) with different cell parameters (detailed structure of Co-bdc and the results in Supplementary Fig. 24). In a similar sense, IRMOF-20, which is an isoreticular structure of MOF-5 but with longer ligand was selected as a potential matching pair with HKUST-1 (Supplementary Fig. 25), and this

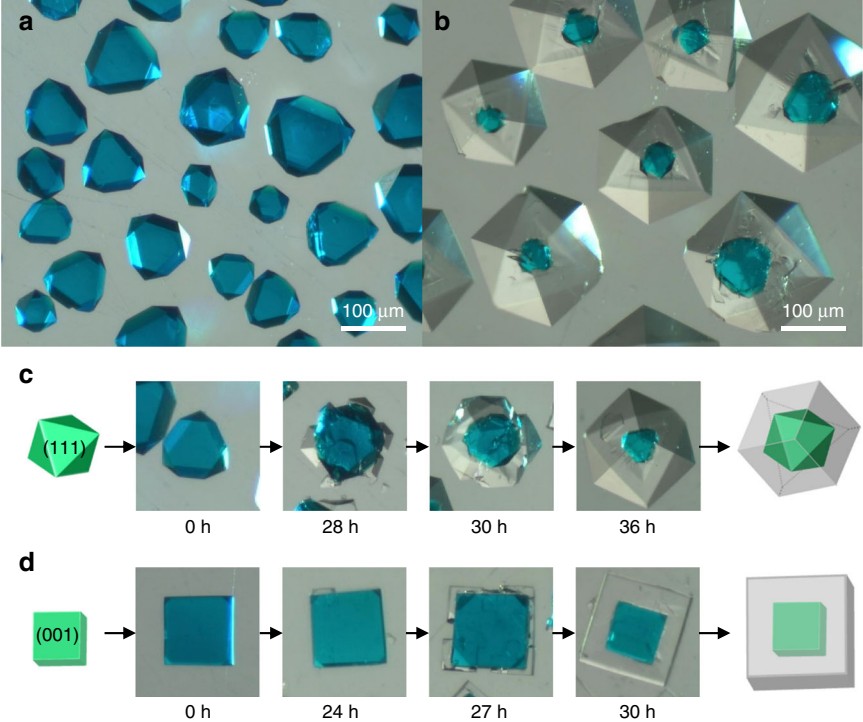

**Fig. 3** Formation of HKUST-1@MOF-5 from the HKUST-1 crystal. **a** Optical microscope image of the octahedral HKUST-1 crystal. **b** Single crystalline HKUST-1@MOF-5. **c** Growth process of the single crystal HKUST-1@MOF-5 from octahedral HKUST-1 for different times. **d** Growth process of single crystalline HKUST-1@MOF-5 from cubic HKUST-1 for different times

pair did not form a single crystalline core-shell structure, with a lot of separate crystals being formed. Although not comprehensive, these examples indicate the importance of designing MOF@MOFs via computational method prior to synthesis to enhance the likelihood of forming a single crystalline domain.

**Other cubic/cubic MOF@MOFs**. To further explore our workflow, we extended to other three cubic/cubic pairs (i.e., HKUST-1@IRMOF-18, UiO-67@HKUST-1, PCN-68@MOF-5) predicted from the screening data as next targets for experimental synthesis (Fig. 4). IRMOF-18 is an isostructural structure with MOF-5 yet composed of tetramethyl terephthalate (bdc-type ligand but more hydrophobic and bulky in nature), and thus this ought to serve as a good counterpart to HKUST-1. The subsequent synthetic reaction of IRMOF-18 on HKUST-1 successfully yielded single crystalline HKUST-1@IRMOF-18 composite (Fig. 4a), similarly to the case for HKSUT-1@MOF-5. As such, functionalization of MOF-5 does not affect the formation of the composite.

As another example, UiO-67@HKUST-1 pair was synthesized and both scanning electron microscopy (SEM) images of UiO-67 and UiO-67@HKUST-1 only showed the octahedral morphology (Supplementary Fig. 26). Thus scanning transmission electron microscopy and energy-dispersive X-ray spectroscopy (STEM-EDS) mapping was employed to reveal the successful synthesis of the core-shell structure, supporting with the XRPD patterns (Fig. 4b and Supplementary Fig. 27).

So far within the successful synthesis of MOF@MOF composites, all chemical connection points of counterpart MOFs were individually well-matched (Supplementary Fig. 28) and possessed the similar lattice parameters. However, in the computationally suggested pair of PCN-68 and MOF-5 (with lattice mismatch of 2.0%), MOF-5 has twice as large chemical connection points per unit area compared to that of PCN-68. Despite the missing linkers of MOF-5, the PCN-68@MOF-5 composite was successfully synthesized, in which each PCN-68

crystal is located within the cubic MOF-5 shell (Fig. 4c and Supplementary Fig. 29).

**Extension to cubic/hexagonal MOF@MOFs**. To further explore the potential of our computational/experimental approach that promotes the oriented growth of MOF crystals, two MOFs with different crystal system (i.e., cubic/hexagonal combination) were selected for potential MOF@MOF composites. Among the cubic/hexagonal candidate pairs extracted from our computational algorithm, the first target system was the pair UiO-66/MIL-88B(Fe) (Supplementary Data 1 and Supplementary Fig. 30). In the simulation study of their crystallographic linking between (111) plane of UiO-66 and (001) of MIL-88B(Fe), the 2D lattice parameters of each other have similar values (14.6374 Å and 14.4162 Å for UiO-66 and MIL-88B(Fe), respectively; 1.5% of mismatch) and their chemical connection points are also well-matched (Supplementary Fig. 31). As a single crystalline 3D substrate for epitaxial growth of MIL-88B, the octahedral UiO-66 crystals were put into the MIL-88B precursor solution and heated at 100 °C for 8 h, resulting in the UiO-66@MIL-88B composites (Fig. 5a, b). Interestingly, the morphology of UiO-66@MIL-88B turns out to be a star-shape crystal resulting from the triangular pyramid MIL-88B crystal grown on the each of the {111} planes of the octahedral UiO-66. XRPD confirms the pure phases of the two MOFs and an EDS mapping indicates a star-shape core-shell structure where Zr atoms exist in the center of particle and Fe atoms, in the outer domains (Fig. 5c and Supplementary Fig. 32). Analyzing the growth behaviors of UiO-66@MIL-88B(Fe) as shown in Fig. 5d, unlike the case of HKUST-1@MOF-5, it can be seen that the triangular pyramid part of MIL-88B(Fe) does not completely cover the entire core UiO-66 structure, even with an extended reaction time of 12 h (Supplementary Fig. 33). One hypothesis that explains this difference is that the two components have different crystal systems and as such, cell matching between the MIL-88B(Fe) islands grown on different (111) planes

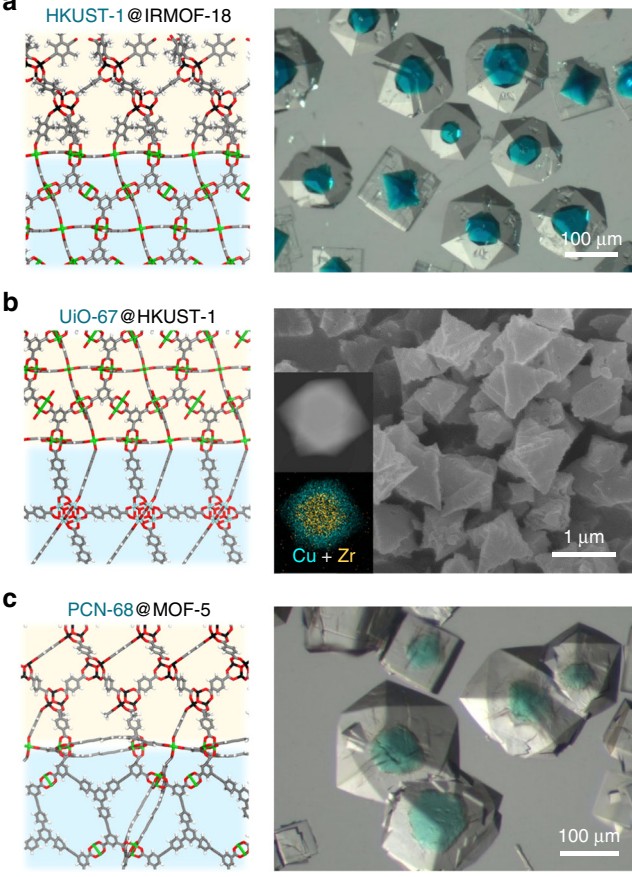

**Fig. 4** Examples of synthesized cubic/cubic MOF@MOFs. **a** Computational structural model from matching the (111) plane and the corresponding optical image of HKUST-1@IRMOF-18. **b** Computational structural model from matching the (111) plane and the SEM image of UiO-67@HKUST-1. Inset is STEM image of UiO-67@HKUST-1 and corresponding overlay mapping of two elements, Zr(orange) and Cu(Cyan), based on EDS. **c** Computational structural model matching the (111) plane and the optical image of PCN-68@MOF-5. In **a**, **b** and **c**, the yellow region indicates the space for IRMOF-18, HKUST-1 and MOF-5, respectively and the green region indicates the space for HKUST-1, UiO-67 and PCN-68, respectively

is not possible, which prevents single crystal growth. This phenomenon provides evidence that the UiO-66 substrate plays a primary role in directing the growth of MIL-88B crystals[42].

Furthermore, UiO-67@MIL-88C(Fe) is another pair obtained from the computational algorithm (0.9% lattice mismatch), whereby elongated ligands were employed to construct the isostructural pair of UiO-66@MIL-88B(Fe). Accordingly, this pair had similar morphology as octapod star-like composites (Supplementary Fig. 34). Here it should be noted that the sequential synthesis of MIL-88A(Fe) and -88C(Fe) on UiO-66 (with the ligands being fumarate and 2,6-naphthalenedicarboxylate, respectively) yielded composites with the same morphological characteristic to that of UiO-66@MIL-88B(Fe), despite relatively unfavorable lattice matching (lattice mismatch = 5.2% for MIL-88A(Fe); 21.9% for MIL-88C(Fe)) (Supplementary Figs. 35–37). According to the previous work conducted by Serre et al., this series of isoreticular analogues MIL-88A to C undergo extensive changes in unit cell due to their large degree of flexibility, (transition from the dried phase to open; Supplementary Table 3)[43]. Since the lattice parameter of UiO-66 in {111} surface does not largely deviate from the extent of these changes in the MIL-88 analogues, the lattice mismatches at the interface between the two connected

MOFs could be compensated by the hinge movements. The current computational algorithm is limiting in this sense as it does not take flexibility into account upon screening for the optimal MOF@MOF pairs. Thus, in the future, we deem this to be an important consideration to further expand the potential matching candidates found in the MOF@MOF structures.

## Discussion

In this work, a joint computational experimental approach was developed to rationally design MOF@MOFs. Our results indicate that the all six pairs predicted from our computational algorithm successfully grew into single crystal MOF@MOF, validating the predictions made from the algorithm. Moreover, the number of predicted pairs can increase even more with a more general 2D lattice matching[44,45] and it is worth investigating in the future. Given that it might be rare where two different MOFs can connect together to form MOF@MOF structures, we devised a strict criterion for matching and as such, it is conceivable that other MOF@MOFs that are outside of our computational predictions can still be experimentally synthesized. Nevertheless, in terms the ability to produce large crystals in a reliable manner, designing these materials from the molecular level using our computational algorithm will most likely lead to more consistent results, which can be important in real-world applications. Finally, we believe that our joint computational/experimental workflow can readily extend into other classes of materials and can lead to rapid exploration of the composite MOFs space for accelerated materials development.

## Methods

**Materials and characterization**. All chemicals and solvents were of reagent grade and were used as received without further purification. Fourier-transform (FT)-NMR spectra were recorded on Agilent 400-MR DD2 spectrometer. XRPD patterns were collected on a Bruker D8 advance diffractometer at 40 kV and 40 mA for Cu K$\alpha$ ($\lambda = 1.54050$ Å), with a step size of 0.02° in 2$\theta$. The nitrogen adsorption–desorption isotherms were obtained using a BELSORP-max at 77 K. Prior to the adsorption measurements, all samples (~100 mg) were evacuated ($p < 10^{-5}$ mbar) at 393 K for 12 h. The specific surface areas were determined from the linear part of the Brunauer-Emmett-Teller (BET) equation. Scanning electron microscopy (SEM) images were taken using a Hitach High-Technologies Cold FE-SEM operating at 10 kV. Scanning electron microscopy/Energy-dispersive X-ray spectroscopy (SEM-EDS) images were taken using a FEI Nova NanoSEM operating at 10 kV. Transmission electron microscopy (TEM), scanning transmission electron microscopy (STEM), and energy-dispersive X-ray spectroscopy (EDS) images were obtained using a FEI Tecnai G2 F20 X-Twin TEM and JEOL JTEM 2100F microscope. The single-crystal diffraction images of HKUST-1, MOF-5, and cubic HKUST-1@MOF-5 were collected with a Rigaku R-Axis Rapid II at room temperature (Mo$_{K\alpha}$, $\lambda = 0.71073$ Å). The crystals were mounted on the tip of a thin glass fiber using epoxy. The Rapid Auto software (R-Axis series, Cat. No. 9220B101, Rigaku Corporation) was used for data collection and processing.

**Synthesis of octahedral HKUST-1**. Octahedral HKUST-1 was prepared by a reported method with minor modifications[41]. Cu(NO$_3$)$_2$·2.5H$_2$O (0.472 g, 2.03 mmol) was dissolved in 6 mL of 1:1 H$_2$O/$N$,$N$-dimethylformamide (DMF) mixture in a 50 mL vial. Benzene-1,3,5-tricarboxylic acid (H$_3$BTC) (0.360 g, 1.71 mmol) was completely dissolved in slightly heated ethanol (4.5 mL) with stirring. To Cu nitrate solution, ethanolic ligand solution and glacial acetic acid (12 mL) were added and placed at 55 °C oven. After 22 h, the mother liquor quickly decanted and the blue crystals were washed with fresh ethanol. For MOF@MOF synthesis, ethanol washed HKUST-1 crystals were stored in $N$,$N$-diethylformamide (DEF) solvent.

**Preparation of octahedral and cubic HKUST-1@MOF-5**. Zn(NO$_3$)$_2$·6H$_2$O (0.760 g, 2.55 mmol) and terephthalic acid (0.132 g, 0.795 mmol) were dissolved in 20 mL of DEF in a glass jar. Five milligrams of filtered octahedral HKUST-1 crystals were added and well dispersed on the bottom of the glass jar. The glass jar was heated at 85 °C. After 36 h, the mother liquor was quickly decanted and HKUST-1@MOF-5 crystals were washed with fresh DEF and dichloromethane. The octahedral HKUST-1@MOF-5 crystal is comprised of bdc to btc ligands in the mole ratio of 12 (based on NMR analysis). The cubic HKUST-1@MOF-5 crystals were prepared with the same procedure except for the use of 5 mg of cubic HKUST-1 crystals instead of octahedral HKUST-1 crystals.

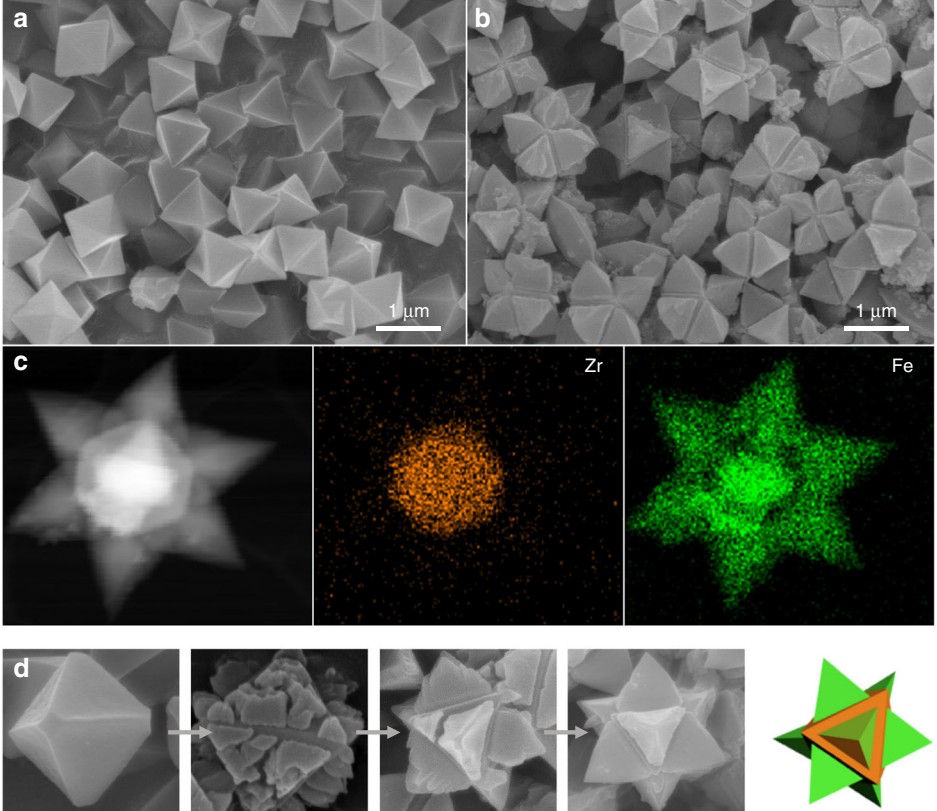

**Fig. 5** Facet-oriented formation of UiO-66@MIL-88B(Fe) from the octahedral UiO-66 crystals. **a** SEM image of octahedral UiO-66 crystals. **b** SEM image of octahedral UiO-66@MIL-88B(Fe) crystals. **c** STEM image of UiO-66@MIL-88B(Fe) and corresponding mapping of two elements, Zr(orange) and Fe (green), based on EDS. **d** Growth process of single crystalline star-shaped UiO-66@MIL-88B(Fe) and schematic image of star-shaped UiO-66@MIL-88B (Fe) single crystal

**Energy calculation using DFT**. DFT calculations were conducted to estimate the stability of interface using Vienna Ab initio Software Package (VASP)[46,47]. Projector augmented wave (PAW) pseudopotentials[48] were used to describe ion-electron interactions of each atom and Perdew, Burke, and Ernzerhof (PBE) exchange-correlation functional[49] was used in all the simulations. A plane wave basis set with energy cutoff of 400 eV was used in all the simulations and a force threshold of 0.02 eV $Å^{-1}$ was used to fully relax the atomic positions. Cluster models were used to reduce the computational cost in this calculations due to complexity of composite MOF@MOF system which have to be represented as large unitcell to lead high computational cost. (Supplementary Figs. 10–11) Thus, only a gamma point was sampled by using $1 \times 1 \times 1$ Monkhorst-Pack k-point grids[50] and unitcells of 30 Å × 30 Å × 30 Å including cluster structures were used.

## Data availability

The data that support the findings of this study are available from the corresponding author upon reasonable request.

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

## Acknowledgements

This work was supported by Samsung Research Funding & Incubation Center of Samsung Electronics under Project Number SRFC-MA1702-07.

## Author contributions

H.R.M. and J.K. formulated the project. O.K. devised the computational algorithm. O.K. and H.P. performed the DFT calculations. J.Y.K., S.P., J.H.L. and J.H. synthesized and characterized the MOFs and MOF@MOF pairs. All authors contributed to writing/revising the paper.

## Additional information

**Competing interests:** The authors declare no competing interests.

