## [Transparent Peer Review File · Nature Communications]

Reviewers' comments:

Reviewer #1 (Remarks to the Author):

The manuscript by Kwon et al. describes a computational study on the lattice match of multiple MOFs with different metal clusters and linkers. They screened thousands of MOFs and selected some typical examples to show that the matched lattice parameters are vital to achieve efficient MOF growth on another MOF. Guided by the calculation results, the authors shown that several single crystalline MOF@MOF composites can be prepared, for example core-shell HKUST-1@MOF-5, and UiO-66@MIL-88B star-like superstructures. This study provides very comprehensive studies on the growth of seamlessly connected MOF composites. I believe it is suitable for publication in Nat. Comm. However, there are some minor points that the authors should address prior to acceptance.

1. In the experiment section, the authors discussed about the island formation and further growth of MOF-5 outside HKUST-1 crystals. This process and the time-dependent growth have been well documented by Feng et al, ACS Cent. Sci., 2018, 4 (12), 1719–1726 (see: Figure 2 and Figure S48). The results should be mentioned, compared and discussed in the main text.

2. It would be interesting to further extend the discussion to some previously reported MOF@MOF structures, for example PCN-222@UiO (Yang et al, Angew. Chem. 2018, 130, 3991). How would the lattice parameters of the PCN and UiO phases contribute to the core-shell growth?

3. The interface is very interesting and different from the previous MOF@MOF report. Did the authors conduct cross-sectional SEM studies or other methods to verify the seamless interface, rather than some coordination polymers at the interface?

4. The references should be improved. For example, when the authors talk about the seamless assembly of MOF superstructures, some recent reports (such as Feng, et al, Chem, 2019, <https://doi.org/10.1016/j.chempr.2019.03.003>) should be added. Although not mentioned in the manuscript, the authors conducted the synthesis of MOF@MOF in a stability-sequence way, for example, studied the growth of relatively labile one on stable MOF seeds. The authors are suggested to provide some discussion on the stability issues (Feng et al, ACS Cent. Sci., 2018, 4 (12), 1719–1726; Yuan et al, Adv. Mater. 2018, 1704303).

5. Since the connection points of MOF-5 and HKUST-1 are matched, is it possible to grow HKUST-1 outsides MOF-5? The discussion should be provided in the manuscript.

Reviewer #2 (Remarks to the Author):

This manuscript describes the theoretical screening of interfacial matching between two different MOFs and the experimental implementation of the synthesis of core/shell MOFs. The authors demonstrated the core/shell fabrication of HKUST-1/MOF-5 in two different ways; either based on the octahedral core (the (111) facet exposed) or based on the cubic core (the (100) facet exposed). This is definitely an interesting result that shows the computational screening discovered the similarity of lattice matches between two MOFs in two different facets and the proof of screening by experimental methods. Further the authors demonstrated several combinations of MOFs for core/shell fabrications. One of the most amazing results is to discover the combination between UiO/MIL series, in general a robust/flexible MOF combination. The manuscript is definitely novel and gives a big impact on the MOF and materials science societies; in particular, the researchers working on materials composites would love this paper. I believe that this manuscript would be potentially accepted in Nature Communications; however, prior to the acceptance, the authors should address the following scientific concerns.

1. I think that the experimental evidence of epitaxial growth is not well provided. The authors nicely describe the interfacial structures between two MOFs by the computational modeling; however, there is no experimental evidence for clarifying the structure. The authors only demonstrated the macroscopic analysis of crystal morphologies and the bulk experiments such as

PXRD and sorption. The detail experimental trial for unveiling the interfacial structure is necessary, in particular for the one combination that grow a large single crystal (HKUST-1/MOF-5). As shown in ref. 10, the interfacial analysis using surface X-ray diffraction setup will be used for evaluating the relationship between two structural features. As shown in the computational modeling, there are two possible interfacial connections for each facet connection, (111) or (100) facets. It would be nice to evaluate this relationship between two structures. I also understand that it is not easy to do such surface XRD experiments. What about using a single crystal diffractometer to evaluate the relationship by separately analyzing two distinct diffraction in reciprocal lattice space?

2. The sorption data does not tell anything about the accessibility to the pores of core HKUST-1. This is because there is no evidence that the core/shell configuration is maintained even after the sorption experiment and the simple physical mixture of HKUST-1 and MOF-5 would give the same data. The authors are highly encouraged to provide the SEM or optical microscopic data after the sorption experiment. If there is a crack in the crystal, the gas can easily diffuse into the core and the authors' hypothesis will be broken. The authors should take highly care of their claims; otherwise, they would lose the scientific significance of this manuscript.

3. It seems that the core crystals were dissolved during the synthesis of core/shell crystals, which is quite obvious for HKUST-1/MOF-5 synthesis (Figs. S 17 and 19). Is this necessary process to show a clear facet for the epitaxial growth? It would be nice to comment on this point.

4. Most importantly, the authors did not provide any experimental evidence of flexibility of MIL-88 system. The results of growing MIL-88A and C on UiO-66 is contradictory to the authors' computational screening and the authors concluded it as its flexibility of MIL-88 series. This would be most plausible explanation but there is no structural evidence provided. The authors should use electron diffraction to clarify the structure of grown shell of MIL-88. At least, the authors should provide and discuss the PXRD data.

5. The paper of facet-selective epitaxial growth of MOFs (Chem. Commun. 2009, 5097.) should be cited in this manuscript because this paper experimentally demonstrated the difference in lattice parameters facilitates the facet-selective growth of a MOF on the top of another MOF surface. This is relevant for the discussion of MIL-88 growth on the surface of UiO-66(67).

Reviewer #3 (Remarks to the Author):

This is an interesting and potentially important manuscript. The growth and control of MOF heterojunctions represents an attractive pathway forward for the field. This study combines theory and experiment to good effect, and reports a wide range of examples to illustrate the point and the generality of the approach. I recommend publication subject to minor changes.

Note that my comments are more focused on the simulation component of the paper.

Suggestions:

* "To the best of our knowledge, this is the first instance where synthesis of composite MOFs originated from computational predictions" - this is a weak line to have in the abstract. It would be better to remove it here. The abstract could also be extended to state some of the positive combinations in order to make it more searchable.

* The screening procedure is fine for the MOFs considered, but more sophisticated algorithms exist that allow for rotations of the lattice vectors to find the best possible match. For example, see Zur and McGill <https://aip.scitation.org/doi/10.1063/1.333084> as applied to ZIFs in <https://pubs.rsc.org/en/content/articlelanding/2017/fd/c7fd00019g#!divAbstract>. This could be

mentioned for future more general extensions of the screening approach.

* The energetics reported for the interfaces are in kJ/mol. It is conventional in the field to present per unit area as this allows for the comparison between different interface models and is what can be physically measured in experiments. There is a recent discussion of different definitions of interface energy definitions that may be of interest (see Section 3) for updating the analysis: <https://iopscience.iop.org/article/10.1088/2515-7655/aad928/meta>. The authors need to be careful when they say the interface is "thermodynamically stable".

Yours sincerely,
Aron Walsh

Below, we have written a point-by-point response to all of the reviewer's comments.

Responses to the Comments of Reviewer #1

- (Reviewer's Comments)** *In the experiment section, the authors discussed about the island formation and further growth of MOF-5 outside HKUST-1 crystals. This process and the time-dependent growth have been well documented by Feng et al, ACS Cent. Sci., 2018, 4 (12), 1719–1726 (see: Figure 2 and Figure S48). The results should be mentioned, compared and discussed in the main text.*

(Authors' Response) We agree that the suggested reference is worth discussing greater detail in our manuscript. Feng *et al.* previously exemplified surface functionalization and retrosynthetic techniques for the preparation of MOF-on-MOF hierarchical structures, in which the HKUST-1@MOF-5 core-shell structure was also synthesized. In the study, the synthetic process was controlled by kinetics to induce the growth of shell MOFs on the core MOFs, while our strategy is based on consideration of the interface connection between the core and shell MOFs. To compare and discuss the reference and our result, we have newly added the following sentences in our revised manuscript.

Changes made :

- We have added the sentence in the Experimental synthesis of HKUST-1@MOF-5 section (page 7) in the revised manuscript.
- "Similar composite has been synthesized in the previous work but achieved via different approach of kinetic control of shell MOF growth³⁶."

36. Feng, F. et al. Uncovering two principles of multivariate hierarchical metal–organic framework synthesis via retrosynthetic design. *ACS Cent. Sci.* **4**, 1719–1726 (2018).

- It would be interesting to further extend the discussion to some previously reported MOF@MOF structures, for example PCN-222@UiO (Yang et al, Angew. Chem. 2018, 130, 3991). How would the lattice parameters of the PCN and UiO phases contribute to the core-shell growth?*

We appreciate the suggestion. In the analysis on the PCN-222@UiO-67, we assumed a cubic/hexagonal pair (admittedly, this information was not available in the paper) with (001) surface of hexagonal PCN-222 and (111) surface of UiO-67. As a result, the lattice mismatch was 9.4%, which was outside of our lattice match criteria. However, even with relaxing the lattice match criterion, the chemical connection points of PCN-222 were not well defined along the (001) plane.

In further analysis, we changed the planes of PCN-222 and UiO-67, and found that relatively well-matched chemical connection points were found along the (111) plane in UiO-67 and the (100) or (010) plane in the PCN-222. The lattice parameters were as follows in these two cases:

PCN-222 (100) plane : $a = 41.968 \text{ \AA}$, $b = 17.143 \text{ \AA}$, angle between two lattice vector : 90°
UiO-67 (111) plane : $a = 32.941 \text{ \AA}$, $b = 19.018 \text{ \AA}$, angle between two lattice vector : 90°

The lattice mismatch along the a axis was 21.5% and along the b axis was -10.9%. Given the relatively large mismatch, a large size of supercell would be required to find well-matched chemical connection points, leading to proportionally smaller number of matching chemical connection points.

As a result, in case of synthesized PCN-222@UiO-67, it is difficult to simply model the well matched interface within long range using our simple model and we anticipate very complex interface between the experimentally synthesized PCN-222@UiO series. One possible scenario is that at first, BPDC linker of UiO-67 is connected on the PCN-222 surface and the seed for UiO-67 growth was locally formed. And then the UiO-67 crystals were grown on the local seeds and merged into poly-crystal. We would like to also emphasize that we do not claim that our algorithm is the only way to find MOF@MOF and as such, there might be some other factors (e.g. flexibility of MOFs) that lead to connection between pairs with mismatched lattice parameters.

3. *The interface is very interesting and different from the previous MOF@MOF report. Did the authors conduct cross-sectional SEM studies or other methods to verify the seamless interface, rather than some coordination polymers at the interface?*

We thank the reviewer's comments. We have conducted the optical microscope measurement in high magnification (at 630 magnification) instead of SEM techniques to exactly judge whether the interface is seamless without electron beam damage. As shown in the figure below, we were able to clearly observe the seamless interface between the HKUST-1 and MOF-5 in a single-crystalline phase. Hence, we have added the following figure as **Supplementary Fig. S13** in the revised manuscript.

Changes made :

- We have revised the sentences in the *Experimental synthesis of HKUST-1@MOF-5* section (page 7) in the revised manuscript.
- "As shown in **Fig. 3b** and **Supplementary Fig. S12**, ~ cubic MOF-5 single crystals." → "As shown in **Fig. 3b** and **Supplementary Fig. S12-13**, ~ cubic MOF-5 single crystals with seamless interfaces."
- We have added the new **Supplementary Fig. S13** in the revised supporting information.

- **Fig. S13**| Optical microscope image of HKUST-1@MOF-5 crystals. Inset image is high magnification optical microscope image at the interface of a HKUST-1@MOF crystal which clearly indicates the seamless interface between the HKUST-1 and MOF-5 crystals.
4. *The references should be improved. For example, when the authors talk about the seamless assembly of MOF superstructures, some recent reports (such as Feng, et al, Chem, 2019, <https://doi.org/10.1016/j.chempr.2019.03.003>) should be added. Although not mentioned in the manuscript, the authors conducted the synthesis of MOF@MOF in a stability-sequence way, for example, studied the growth of relatively labile one on stable MOF seeds. The authors are suggested to provide some discussion on the stability issues (Feng et al, ACS Cent. Sci., 2018, 4 (12), 1719–1726; Yuan et al, Adv. Mater. 2018, 1704303).*

Following the reviewer's suggestion, we have cited the suggested references. As the reviewer pointed out, the stability of MOFs is a very important issue for synthesizing single-crystalline core-shell MOFs, even though the initial part of our workflow mainly focus on the interface configuration between the core and shell MOFs. Therefore, we have clarified the stability issue with the newly added sentences and references.

Changes made :

- We have revised the sentence in the page 2 in the revised manuscript.
- "More recently, Feng et al. ~ analysis³⁶." → "More recently, Zhou group reported several interesting works for synthesis of hierarchical MOFs that are composed of two different MOFs by kinetic control such as surface functionalization and temperature adjustment^{15, 36-37}."

37. Feng, L., Li, J-L., Day, G. S., Lv, X-L. & Zhou, H.-C. Temperature-controlled evolution of nanoporous MOF crystallites into hierarchically porous superstructures. *Chem* **5**, 1265-1274 (2019).

- We have added the sentence in the Experimental synthesis of HKUST-1@MOF-5 section (page 7) in the revised manuscript.
- "due to their stronger crystal stability against the sequential solvothermal reactions (Fig. 3a) and they were heated at ~" → "due to their stronger crystal stability against the sequential solvothermal reactions (Fig. 3a), which also has been exploited in other MOF@MOF syntheses.^{36, 40} They were heated at ~"

36. Feng, F. et al. Uncovering two principles of multivariate hierarchical metal–organic framework synthesis via retrosynthetic design. *ACS Cent. Sci.* **4**, 1719–1726 (2018).

40. Yuan, S. et al. Stable metal-organic frameworks: design, synthesis, and applications. *Adv. Mater.* 1704303 (2018).

5. *Since the connection points of MOF-5 and HKUST-1 are matched, is it possible to grow HKUST-1 outsides MOF-5? The discussion should be provided in the manuscript.*

According to our simulation results, HKUST-1 would be able to reversely grow on the MOF-5 crystal. However, in the actual synthesis, the stability of core MOFs is very critical to retain its integrity during sequential solvothermal reactions. As shown in **Reviewer only Fig. 1**, MOF-5 crystals lose their crystallinity not only in the original HKUST-1 synthetic condition (HKUST-1 precursor dissolved in 1:1:1 mixture of DMF:ethanol:H₂O), but also in a modified condition (HKUST-1 precursor dissolved in pure DMF). As a result, we failed to obtain MOF-5@HKUST-1 core-shell structures by using MOF-5 as a core. For clarity, we have reflected this point in revised manuscript.

Reviewer only Fig. 1 | **a**, Optical microscopic images MOF-5 crystals in original HKUST-1 synthetic condition (HKUST-1 precursor dissolved in 1:1:1 mixture of DMF:ethanol:H₂O). | **b**, Optical microscopic images MOF-5 crystals in modified HKUST-1 synthetic condition (HKUST-1 precursor dissolved in pure DMF).

Changes made :

- We have added the sentence in the Experimental synthesis of HKUST-1@MOF-5 section (page 7) in the revised manuscript.
- “due to their stronger crystal stability against the sequential solvothermal reactions (Fig. 3a) and they were heated at ~” → “due to their stronger crystal stability against the sequential solvothermal reactions (Fig. 3a), which also has been exploited in other MOF@MOF syntheses.^{36,40} They were heated at ~”

36. Feng, F. et al. Uncovering two principles of multivariate hierarchical metal–organic framework synthesis via retrosynthetic design. *ACS Cent. Sci.* **4**, 1719–1726 (2018).

40. Yuan, S. et al. Stable metal-organic frameworks: design, synthesis, and applications. *Adv. Mater.* 1704303 (2018).

Responses to the Comments of Reviewer #2

1. *I think that the experimental evidence of epitaxial growth is not well provided. The authors nicely describe the interfacial structures between two MOFs by the computational modeling; however, there is no experimental evidence for clarifying the structure. The authors only demonstrated the macroscopic analysis of crystal morphologies and the bulk experiments such as PXRD and sorption. The detail experimental trial for unveiling the interfacial structure is necessary, in particular for the one combination that grow a large single crystal (HKUST-1/MOF-5). As shown in ref. 10, the interfacial analysis using surface X-ray diffraction setup will be used for evaluating the relationship between two structural features. As shown in the computational modeling, there are two possible interfacial connections for each facet connection, (111) or (100) facets. It would be nice to evaluate this relationship between two structures. I also understand that it is not easy to do such surface XRD experiments. What about using a single crystal diffractometer to evaluate the relationship by separately analyzing two distinct diffraction in reciprocal lattice space?*

We are grateful to the reviewer's insightful comment. As the reviewer mentioned, it is difficult for us to conduct such surface XRD experiments under our present circumstances, so we have exploited a single-crystal diffractometer to further prove the epitaxial growth. In this experiment, we examined the cubic shape HKUST-1@MOF-5 crystal due to its clearly distinguishable crystal facets, which was measured by 'R-axis RAPID II' equipment of the Rigaku Corporation.

As shown in Revised Fig. S22, the diffraction images for HKUST-1, MOF-5, and HKUST-1@MOF-5 were obtained by incident X-ray beam perpendicular to a {100} plane (i.e. perpendicular to the face of the cube) with the help of the program. Note that the X-ray beam size is large enough to cover the whole core-shell sample (500 μm x 500 μm), where the diffraction image of the core-shell crystal should contain the reflections from both the crystals, core HKUST-1 and shell MOF-5. The three diffraction images are 4-fold symmetric with characteristic reflections of {0 2 0}, {2 2 0}, {12 2 0}, {0 12 0}, and {10 6 0}, which implies that the {100} plane is perpendicularly well-aligned to the incident X-ray beam. Since MOF-5 and HKUST-1 have the same crystal system and similar cell parameters, the most of diffraction spots appears at the same positions, as shown in **Supplementary Fig. S22 b** and **c**. The diffraction image of the cubic HKUST-1@MOF-5 crystal is exactly overlapped one of each diffraction for HKUST-1 and MOF-5 without any splitting and twinning spots (**Supplementary Fig. S22d**). It implies that cubic HKUST-1@MOF-5 crystal is built from the epitaxial growth with good crystal orientation matches.

To double-check this point, we further obtained the diffraction images for the three samples while rotating 180° on the ω -axis (Revised Fig. S23), and then extracted the intensity profiles on the same 2θ . The reflections within the 2θ range of $4.4\pm 0.2^\circ$ and $6.2\pm 0.2^\circ$ represent the planes {2 2 0} and {4 0 0}, respectively. For the HKUST-1@MOF-5, the periodic peaks without any splitting indicate that the core crystal and shell crystal are well-arrayed along the ω -axis. Hence, we confirmed the epitaxial growth in HKUST-1@MOF-5 crystal and added the following figures as Fig. S22 and S23 in the revised supporting information.

Changes made :

- We mentioned the newly added figures (Supplementary Fig. S22 and S23) in the related sentence in the revised manuscript (*page 8*): thereby creating the cubic HKUST-1@MOF-5 structure in the single crystal phase (**Fig. 3d** and **Supplementary Fig. S20-S23**).
- We have added the sentence in the *Methods section* (*page 13*) in the revised manuscript.

- “The single-crystal diffraction images of HKUST-1, MOF-5, and cubic HKUST-1@MOF-5 were collected with a Rigaku R-Axis Rapid II at room temperature (M_{OxK_α} , $\lambda = 0.71073 \text{ \AA}$). The crystals were mounted on the tip of a thin glass fiber using epoxy. The Rapid Auto software (R-Axis series, Cat. No. 9220B101, Rigaku Corporation) was used for data collection and processing.”
- We have added the new **Supplementary Fig. S22** and **S23** in the revised supporting information.

Fig. S22 | a, Schematic illustration of the measurement of the diffraction images and simulated X-ray diffraction image. X-ray diffraction images for cubic shape HKUST-1, MOF-5 and HKUST-1@MOF-5, which were obtained by incident X-ray beam perpendicular to the $\{100\}$ plane with 1° of ω -axis oscillation. **b**, HKUST-1. **c**, MOF-5. **d**, HKUST-1@MOF-5. Note that the (h k l) of diffraction peaks assigned when the X-ray beam is incident along the [100] direction. A simulated X-ray diffraction image was generated under the condition that the incident X-ray beam is perpendicular to the (100) plane of MOF-5 using a CrystalMaker[®]: CrystalMaker Software Ltd, Oxford, England (www.crystallmaker.com).

The diffraction images are 4-fold symmetric with characteristic reflections of $\{0\ 2\ 0\}$, $\{0\ 2\ 2\}$,

$\{0\ 2\ 12\}$, $\{0\ 12\ 0\}$, and $\{0\ 6\ 10\}$, which implies that they all are well-aligned to the $[100]$ direction along the X-ray beam. Thus, the results have demonstrated that the cubic HKUST-1@MOF-5 is built from the epitaxial growth. Note that the X-ray beam size ($500\ \mu\text{m} \times 500\ \mu\text{m}$) is large enough to cover the whole sample, where the diffraction image of the core-shell must include the reflections of both the crystals, core HKUST-1 and shell MOF-5.

Fig. S23 | **a**, Schematic illustration of the measurement of the diffraction images rotating 180° on the ω -axis and X-ray diffraction image. **b**, Crystal images of HKUST-1, MOF-5, and HKUST-1@MOF-5 before the measurement. **c**, Corresponding diffraction images measured

while rotating 180° on the ω -axis. **d**, Intensity profiles obtained by the intensity integration per the β angle within the same 2θ range of $4.4\pm 0.2^\circ$, i.e. $\{h\ k\ l\} = \{2\ 2\ 0\}$. **e**, Intensity profiles obtained by the intensity integration per the β angle within the same 2θ range of $6.2\pm 0.2^\circ$, i.e. $\{h\ k\ l\} = \{4\ 0\ 0\}$.

For the HKUST-1@MOF-5, the periodic peaks without any splitting indicate that the core crystal and shell crystal are arrayed along the ω -axis.

- The sorption data does not tell anything about the accessibility to the pores of core HKUST-1. This is because there is no evidence that the core/shell configuration is maintained even after the sorption experiment and the simple physical mixture of HKUST-1 and MOF-5 would give the same data. The authors are highly encouraged to provide the SEM or optical microscopic data after the sorption experiment. If there is a crack in the crystal, the gas can easily diffuse into the core and the authors' hypothesis will be broken. The authors should take highly care of their claims; otherwise, they would lose the scientific significance of this manuscript.*

We thank the reviewer for bringing this issue. In the original manuscript, in order to confirm the accessibility of small molecules to the core HKUST-1, we have conducted the solvent exchange and gas sorption experiments. Following the reviewer's comments, we further conducted optical microscope measurements of core-shell crystals in both experiments. In the solvent exchange of diethylformamide (DEF) with dichloromethane, the color change of HKUST-1 core crystals (green to dark blue) occurred in the single-crystalline core-shell crystals, indicating the accessibility of guest molecules into the core HKUST-1 crystal (Revised Supplementary Fig. S17). However, after activation of the core-shell crystal for N_2 sorption measurement, the optical microscope image showed the formation of cracks in the shell crystals, which can induce the facile diffusion of gas molecules into the HKUST-1 core crystal through the cracks (Reviewer only figure 2). As a result, our hypothesis can be supported only by the solvent exchange experiment. Hence, we properly revised the discussion part for the gas sorption experiment in our revised manuscript.

Revised Fig. S17 | **a**, Photographs images and corresponding optical microscope images of HKUST-1@MOF-5 crystals in diethylformamide (DEF). **b**, Photographs images and corresponding optical microscope images of HKUST-1@MOF-5 crystals in dichloromethane (CH₂Cl₂). Strong color change of HKUST-1 in HKUST-1@MOF-5 crystals depending on solvents indicates the accessible and interconnected pore space at the interface between HKUST-1 and MOF-5.

Reviewer Only Fig.2 | **a**, Photographs images of HKUST-1@MOF-5 crystals in glass sample cell after activation. **b**, Optical microscope image of HKUST-1@MOF-5 crystals in glass sample cell after activation.

Changes made :

- We have revised the sentences in the *Experimental section* (page 8) in the revised manuscript.
 - “It should be noted that ~ (**supplementary Fig. S15-16**).” → “The HKUST-1@MOF-5 crystal has a reasonable surface area of 3356 m²/g, considering the surface area values for each of the MOFs (i.e. 2021 m²/g for HKUST-1 and 3598 m²/g for MOF-5) (**Supplementary Fig. S16**). It should be noted that the color change of HKUST-1 core crystal (green to dark blue) occurs upon the solvent exchange of diethylformamide (DEF) into dichloromethane (**Supplementary Fig. S17**).”
 - We have revised the **Supplementary Fig. S15** in original supporting information into **Revised Supplementary Fig. S17** in revised Supporting information.
3. *It seems that the core crystals were dissolved during the synthesis of core/shell crystals, which is quite obvious for HKUST-1/MOF-5 synthesis (Figs. S 17 and 19). Is this necessary process to show a clear facet for the epitaxial growth? It would be nice to comment on this point.*

We thank the reviewer’s comments. As reviewer pointed out, it is possible that some of the HKUST-1 crystal surfaces are dissolved during the core-shell synthesis. However, even if it happens, the same crystal plane inside is exposed leading to epitaxial growth of MOF-5.

4. *Most importantly, the authors did not provide any experimental evidence of flexibility of MIL-88 system. The results of growing MIL-88A and C on UiO-66 is contradictory to the authors’*

computational screening and the authors concluded it as its flexibility of MIL-88 series. This would be most plausible explanation but there is no structural evidence provided. The authors should use electron diffraction to clarify the structure of grown shell of MIL-88. At least, the authors should provide and discuss the PXRD data.

We are grateful to the reviewer's valuable comment. As the reviewer mentioned, selected area electron diffraction (SAED) can be an applicable tool to analyze the structure of the shell MIL-88. However, according to the previous literatures, measuring flexible MOFs with TEM equipment would be very challenging as it requires sophisticated analytic techniques, which we think beyond the scope of our present study. Instead, to dispel the reviewer's concern, we have discussed the XRPD data in the revised manuscript with newly adding Fig. S36 and S37. The XRPD results confirm the pure syntheses of the two MOFs for UiO-66@MIL-88A(Fe) as well as UiO-66@MIL-88C(Fe), which also implies that their (001) planes are compatible with (111) plane of UiO-66. Notably, two different dynamic phases of MIL-88C(Fe) coexist in UiO-66@MIL-88C(Fe).

Changes made :

- We have added the new **Supplementary Fig. S36** and **S37** in the revised supporting information.

Fig. S36 | XRPD patterns of UiO-66(blue) and UiO-66@MIL-88A(Fe) (red) with the simulated XRPD patterns from single crystal data from UiO-66 (black) and MIL-88A(Fe) (gray).

Fig. S37 | **a**, XRPD patterns of UiO-66(blue), MIL-88C(Fe) (light blue), and UiO-66@MIL-88C(Fe) (red) with the simulated XRPD patterns from single crystal data from UiO-66 (black) and MIL-88C(Fe) (gray). Note that the peaks marked with an asterisk result from a different phase of MIL-88C(Fe).²² **b**, XRPD patterns of UiO-66 (blue) and activated UiO-66@MIL-88C(Fe) (red) with the simulated XRPD patterns from single crystal data from UiO-66 (black) and activated MIL-88C(Fe) (gray), which indicate that the MIL-88C(Fe) grown on UiO-66 is the pure phase.

5. *The paper of facet-selective epitaxial growth of MOFs (Chem. Commun. 2009, 5097.) should be cited in this manuscript because this paper experimentally demonstrated the difference in lattice parameters facilitates the facet-selective growth of a MOF on the top of another MOF surface. This is relevant for the discussion of MIL-88 growth on the surface of UiO-66(67).*

We do agree that the suggested reference will reinforce our results. Thus, following the reviewer comment, we newly added the suggested reference in revised manuscript.

Changes made :

- We have newly added the suggested reference in revised manuscript.
- “This phenomenon provides evidence that the UiO-66 substrate plays a primary role in directing the growth of MIL-88B crystals.⁴²”

42. Furukawa, S. et al. A block PCP crystal: anisotropic hybridization of porous coordination polymers by face-selective epitaxial growth. *Chem. Commun.* 5097–5099 (2009).

Responses to the Comments of Reviewer #3

1. *"To the best of our knowledge, this is the first instance where synthesis of composite MOFs originated from computational predictions" - this is a weak line to have in the abstract. It would be better to remove it here. The abstract could also be extended to state some of the positive combinations in order to make it more searchable.*

Thank you for your comment about abstract. We have removed the line and have modified a line about the pair we synthesized in our abstract. (page 1)

Changes made :

- "Six MOF pairs (HKUST-1@MOF-5, HKUST-1@IRMOF-18, UiO-67@HKUST-1, PCN-68@MOF-5, UiO-66@MIL-88B(Fe) and UiO-67@MIL-88C(Fe)) yielded from our theoretical predictions were successfully synthesized, leading to clean single crystalline MOF@MOF, demonstrating the power of our joint workflow."
 - "To the best of our knowledge, this is the first instance of composite MOFs originated from computational predictions, and as such our work can serve as a starting point..." → "Our work can serve as a starting point..."
2. *The screening procedure is fine for the MOFs considered, but more sophisticated algorithms exist that allow for rotations of the lattice vectors to find the best possible match. For example, see Zur and McGill <https://aip.scitation.org/doi/10.1063/1.333084> as applied to ZIFs in <https://pubs.rsc.org/en/content/articlelanding/2017/fd/c7fd00019g#!divAbstract>. This could be mentioned for future more general extensions of the screening approach.*

Thank you for your suggestion. We have added the sentences and suggested reference in discussion section (page 12).

Changes made :

- "Moreover, the number of predicted pairs can increase even more with a more general 2D lattice matching^{44,45} and it is worth investigating in the future. "
44. A. Zur, and T. C. McGill. Lattice match: An application to heteroepitaxy. *J. Appl. Phys.* **55**, 378 (1984)
45. Keith T. Butler et al. Designing porous electronic thin-film devices: band offsets and heteroepitaxy. *Faraday Discuss.*, **201**, 207 (2017)
3. *The energetics reported for the interfaces are in kJ/mol. It is conventional in the field to present per unit area as this allows for the comparison between different interace models and is what can be physically measured in experiments. There is a recent discussion of different definitions of interface energy definitions that may be of interest (see Section 3) for updating the analysis: <https://iopscience.iop.org/article/10.1088/2515-7655/aad928/meta>. The authors need to be careful when they say the interface is "thermodynamically stable".*

Thank you for your suggestion about energy analysis of interface. We have changed the units from kJ/mol to eV/Å² similar to how it is mentioned in the paper (<https://iopscience.iop.org/article/10.1088/2515-7655/aad928/meta>) and modified few

sentence in accordance to your comments. (page 7)

Changes made :

- The values we calculated using DFT were changed in manuscript and supplementary information into proper value with $\text{eV}/\text{\AA}^2$.
- "As a result, the net energy stabilization was $-0.0142 \text{ eV}/\text{\AA}^2$ at the (111) surface, making the MOF@MOF thermodynamically stable" → "As a result, the net energy stabilization was $-0.0142 \text{ eV}/\text{\AA}^2$ at the (111) surface."
- "Given that both of these structures were deemed to be thermodynamically stable, the HKUST-1/MOF-5 system was targeted for the initial experimental synthesis." → "Given the relatively low energies of these structures, the HKUST-1/MOF-5 system was targeted for the initial experimental synthesis."

REVIEWERS' COMMENTS:

Reviewer #1 (Remarks to the Author):

The authors have addressed all comments and I recommend its publication on Nature Comm.

Reviewer #2 (Remarks to the Author):

This revised manuscript definitely improved the quality of the research. In particular, the authors paid more attention to the interfacial structural determinations at the molecular level. This evaluation is mandatory to address the issue of interfacial relationship between the core and shell crystals. I believe that this revised version is good enough to be published as it is.

Shuhei Furukawa

Below, we have written a point-by-point response to all of the reviewer's comments.

Responses to the Comments of Reviewer #1

1. **(Reviewer's Comments)** *The authors have addressed all comments and I recommend its publication on Nature Comm.*

(Authors' Response) Thank you for your comments.

Responses to the Comments of Reviewer #2

1. *This revised manuscript definitely improved the quality of the research. In particular, the authors paid more attention to the interfacial structural determinations at the molecular level. This evaluation is mandatory to address the issue of interfacial relationship between the core and shell crystals. I believe that this revised version is good enough to be published as it is.*

Shuhei Furukawa

(Authors' Response) Thank you for your comments.